# A Pilot Study of Transbronchial Biopsy Using Endobronchial Ultrasonography with a Guide Sheath in the Diagnosis of Peripheral Pulmonary Lesions in Patients with Interstitial Lung Disease

**DOI:** 10.3390/diagnostics11122269

**Published:** 2021-12-04

**Authors:** Takayasu Ito, Tomoki Kimura, Kensuke Kataoka, Shotaro Okachi, Keiko Wakahara, Naozumi Hashimoto, Yasuhiro Kondoh

**Affiliations:** 1Department of Respiratory Medicine and Allergy, Tosei General Hospital, Seto 489-8642, Japan; takaito9@med.nagoya-u.ac.jp (T.I.); kataoka@tosei.or.jp (K.K.); kondoh@tosei.or.jp (Y.K.); 2Department of Respiratory Medicine, Nagoya University Graduate School of Medicine, Nagoya 466-8560, Japan; s.okachi@med.nagoya-u.ac.jp (S.O.); wakahara@med.nagoya-u.ac.jp (K.W.); hashinao@med.nagoya-u.ac.jp (N.H.)

**Keywords:** bronchoscopy, endosonograph, image-guided biopsy, interstitial lung diseases, lung neoplasms

## Abstract

The occurrence of interstitial lung disease (ILD) with peripheral pulmonary lesions (PPLs) is closely linked to the development of lung cancer. Yet, the best diagnostic approach for identifying PPLs in patients with ILD remains elusive. This study retrospectively investigated the application of transbronchial biopsy (TBB) using endobronchial ultrasonography with a guide sheath (EBUS-GS) to the effective and safe diagnosis of PPLs when compared with conventional TBB. The study sample included a consecutive series of 19 patients with ILD who underwent conventional TBB or TBB using EBUS-GS at Tosei General Hospital between 1 April 2013 and 31 October 2015. The two techniques were compared based on diagnostic yield and associated complications. The diagnostic yield of EBUS-GS TBB was significantly higher than that of conventional TBB (*p* = 0.009), especially for small lesions (≤20 mm), lesions located in the lower lobes, lesions with a positive bronchus sign, and lesions visible by chest radiography (*p* = 0.010, *p* = 0.022, *p* = 0.006, and *p* = 0.002, respectively). There were no significant differences in complication rates. Therefore, EBUS-GS is an effective alternative for the diagnosis of PPLs in patients with ILD, without additional complications.

## 1. Introduction

The widespread use of low-dose computed tomography (CT) for lung cancer screening has produced an increase in the detection rates of peripheral pulmonary lesions (PPLs) [1]. In the presence or absence of prior clinical symptoms, identification of interstitial lung disease (ILD) is often incidental with PPL detection. Furthermore, new PPLs frequently appear during follow-up of ILD, and concomitant lung cancer is common, especially in patients with idiopathic pulmonary fibrosis (IPF) [2]. Standard approaches to the evaluation of PPLs include bronchoscopy, transthoracic needle biopsy (TTNB), and surgical lung biopsy (SLB). However, the diagnosis of PPLs using a conventional flexible bronchoscope presents technical challenges that impact diagnostic certainty. The diagnostic yield of bronchoscopy varies according to PPL size and location, and the sensitivity ranges from 14% to 63% [3]. In the last few decades, complementing bronchoscopy with radial endobronchial ultrasound (R-EBUS) and navigation modalities has strengthened the reliability of biopsy sample collection and improved the diagnostic yield of PPLs [3,4,5,6]. Consequently, the technique of endobronchial ultrasonography with a guide sheath (EBUS-GS) and an adjunct navigation system is fast becoming the leading procedure for the diagnosis of PPLs and is associated with fewer reported complications when compared with TTNB or SLB [3,5].

However, there are few reports detailing the application of EBUS-GS in patients with ILD, specifically. In patients with ILD, TTNB is associated with a higher frequency of pneumothorax, and SLB carries the risk of acute ILD exacerbation [7]. In addition to safety concerns, diagnostic approaches using conventional transbronchial biopsy (TBB) may be less effective in ILD-afflicted lungs because reticular shadows around PPLs preclude their detection and make it difficult to perform biopsies from the lesions appropriately. We hypothesized that the diagnostic yield of PPLs in patients with ILD would be improved using EBUS-GS compared with conventional TBB. The aim of this study was to retrospectively evaluate and compare the diagnostic yields and safety profiles of TBB using EBUS-GS and conventional TBB, when applied to PPL diagnosis in patients with ILD.

## 2. Materials and Methods

### 2.1. Patients

We reviewed the medical records of 38 consecutive patients who underwent either conventional TBB (*n* = 19) or TBB using EBUS-GS (*n* = 19) for PPL diagnosis at Tosei General Hospital between 1 April 2013 and 31 October 2015. The patients were divided into two groups according to the biopsy technique: a conventional TBB group and an EBUS-GS group. PPLs were defined as visible lesions surrounded by normal lung parenchyma or interstitial lung lesions that were identified by CT but not visualized by bronchoscopy. The inclusion criteria were as follows: confirmed ILD accompanied by bilateral reticular shadows on chest radiographs and the application of conventional TBB and TBB using EBUS-GS for PPL sample recovery and diagnosis. Patients with endobronchial lesions and uncertain diagnoses were excluded. ILD was identified by high-resolution CT and clinical data collected before bronchoscopy [8]. Written informed consent was obtained prior to undergoing bronchoscopy from all patients. This study was approved by the Tosei General Hospital Institutional Review Board (IRB#960).

### 2.2. Bronchoscopy Procedure

In most patients, the pulmonary function test (PFT) (spirometry and diffusing capacity of the lung for carbon monoxide) was performed within the month leading up to bronchoscopy. Before conducting either biopsy technique, the upper airway was anesthetized with a 2% lidocaine spray, and an intravenous bolus of midazolam was administered.

A flexible bronchoscope (BF-P260F; Olympus Corporation, Tokyo, Japan) was used in both the conventional TBB and EBUS-GS groups. In the conventional TBB group, the biopsy was performed under fluoroscopic guidance. In contrast, a bronchoscope and a guide sheath (K-201; Olympus Corporation, Tokyo, Japan) were used with a 1.4-mm R-EBUS probe (UM-S20-17S; Olympus Corporation, Tokyo, Japan) in the EBUS-GS group. Radial probe position was categorized as follows: within (the probe was located inside the PPL), adjacent to (the probe was located at the periphery of the PPL), and outside (the probe was located away from the PPL) [4]. Based on helical CT data, a virtual bronchoscopy navigation system (Bf-NAVI; Cybernet Systems, Tokyo, Japan) with a 0.5-mm slice width was used at the physician’s discretion.

### 2.3. Variables

The following clinical information was collected from the records of all included patients: age, sex, underlying ILD (IPF/non-IPF), PFT results, lesion size, lesion structure, lesion lobe, lesion location from the hilum, bronchus sign, visibility on chest radiograph, EBUS image, sampling number, examination time, bronchoscopy diagnosis, and final diagnosis. The structure of the lesion was classified into two groups as follows: solid or others (part-solid or pure ground-glass) [9]. The location of the lesion relative to the hilum was classified into two groups as follows: inner (lesions in the inner- and middle-third ellipses) and outer (lesions in the outer-third ellipse) [10]. The bronchus sign on CT refers to the relationship between the nearest bronchus and lesion.

The success of the final pathological diagnosis was assessed based on the histopathology and/or microbiological analyses of biopsy tissue obtained by bronchoscopy, TTNB, or SLB, as well as clinical follow-up. Successful diagnosis of malignant lesions using bronchoscopy was defined by the recovery of tissue from target lesions that was later confirmed as malignant by histopathology. In contrast, a failed diagnosis was defined by inaccurate or inadequate sample collection (e.g., peripheral lung tissue or peribronchial tissue) and termed non-diagnostic. Bronchoscopic diagnosis of benign lesions was defined as successful depending on whether a definitive diagnosis was obtained histologically and/or microbiologically based on the bronchoscopy-based findings and clinical features. Successful diagnosis of benign lesions using bronchoscopy was defined when the specific findings (e.g., granuloma) were confirmed via bronchoscopy and the lesions regressed spontaneously over the follow-up period; the lesions were then diagnosed as inflammatory lesions. Benign lesions, which could not be pathologically or microbiologically diagnosed after performing bronchoscopy, were evaluated by confirming radiologic size stability over the follow-up period (at least two years) after bronchoscopy. Bronchoscopy was defined as having failed in the case of such benign lesions.

### 2.4. Statistical Analyses

Data were presented as counts, medians, and ranges. Mann–Whitney U tests and Pearson chi-squared tests were used to analyze continuous and categorical variables, respectively. Statistical significance was set at *p* < 0.05, and all reported *p*-values were two-sided. All analyses were performed using the SPSS Statistics software version 28 (IBM, Armonk, NY, USA).

## 3. Results

During the study period, biopsy was performed for one PPL each in 19 patients with ILD who underwent TBB, as well as in another 19 patients with ILD who underwent EBUS-GS (Figure 1). The patient and PPL characteristics are summarized in Table 1. There were no significant differences between the groups in percent predicted forced vital capacity, percent predicted diffusing capacity of the lung for carbon monoxide, lesion size, lesion structure, lobe, location, bronchus sign, and radiographic visibility.

Table 2 summarizes the diagnostic yields of the two techniques. Overall, the diagnostic yield of the EBUS-GS TBB group was significantly higher than that of the conventional TBB group (63.2% vs. 21.1%, *p* = 0.009). EBUS-GS was more effective in diagnosing smaller PPLs (≤20 mm; 71.4% vs. 0%, *p* = 0.010), although there was no significant difference between the methods for detecting lesions > 20 mm. Moreover, EBUS-GS returned significantly higher diagnostic yields for lesions located in the lower lobes, those with a positive bronchus sign, and lesions visible by chest radiography (*p* = 0.022, *p* = 0.006, *p* = 0.002, respectively).

The diagnostic yield of the EBUS-GS group did not vary significantly from that of the conventional TBB group according to the probe position (within, 63%; adjacent to, 60%; and outside, 33%, respectively; *p* = 0.39).

The histological findings of the TBB versus the EBUS-GS group are summarized in Table 3. Squamous cell carcinoma was the most frequently identified malignant lesion using either technique.

There were no significant differences in complication rates between the two groups (Table 4). One of the 19 patients (5.3%) who underwent EBUS-GS experienced a pneumothorax that did not require chest drainage. Two of the 19 patients (10.5%) who underwent conventional TBB experienced mild bleeding, and hemostasis was confirmed only by the bronchial wedge.

## 4. Discussion

To the best of our knowledge, this is the first study to comparatively investigate the efficacy and safety of TBB using EBUS-GS and conventional TBB when diagnosing PPLs in patients with ILD. We hypothesized that reticular shadows around the PPLs in patients with ILD would make their detection challenging, resulting in a lower diagnostic yield. In our findings, the alternative use of EBUS-GS contributed to the improved diagnosis of small PPLs (≤20 mm), lesions located in the lower lobes, lesions with a positive bronchus sign, and visible lesions on chest radiographs.

In this study, the diagnostic yield of small PPLs (≤20 mm) in patients with ILD using conventional TBB was comparatively lower than previously reported diagnostic yields in a variety of patients, not limited to those with ILD [3]. Small PPLs are often difficult to detect. In patients with ILD, reticular shadows compound the challenge of detecting small PPLs. Herth and colleagues [11] reported bronchoscopy with R-EBUS to be an effective procedure for diagnosing small lesions that are invisible on fluoroscopy. Our findings confirm EBUS-GS to be effective for diagnosing small PPLs in patients with ILD as well.

In our study, EBUS-GS improved the diagnostic yield of lesions located in the lower lobe when compared with conventional TBB. The approach to a lesion located in the lower lobe is influenced by respiratory changes rendering repeated TBBs difficult to perform. The use of EBUS-GS offers an effective alternative for repeated biopsies from PPLs. Further, EBUS-GS allows verification of positioning in the target lesion and allows this position to be maintained (Figure 2) [4].

It has been reported that intra-procedural EBUS visualization was a significant factor for achieving successful diagnosis. Moreover, a positive bronchus sign has been shown to correlate with EBUS visualization [12,13,14,15]. We found that the diagnostic yield for lesions with a positive bronchus sign was higher by EBUS-GS than by conventional TBB. Even when conventional TBB was performed to diagnose lesions with a positive bronchus sign, whether the device reached the lesion appropriately by means of conventional TBB or not remains unknown under fluoroscopy. We considered that the diagnostic strength of EBUS-GS for lesions with a positive bronchus sign is accredited to the dual effect of being able to appropriately reach the target lesion and providing the opportunity for repeated biopsies.

In the patients with ILD, approximately 20% of the PPLs biopsied by conventional TBB and EBUS-GS were invisible by chest radiography. There was no significant difference in the diagnostic yield for lesions invisible on radiographs between the two groups; however, we acknowledge that our observation may have been affected by sample size. For lesions detected during the chest X-ray (approximately 80%), the diagnostic yield of EBUS-GS was significantly higher than that of conventional TBB, which is considered to be in agreement with the explanation that whether the lesion was visible on chest X-ray or not depends on whether the device was guided to lesions appropriately during EBUS-GS [4]. Therefore, EBUS-GS TBB was considered more effective for diagnosing lesions detected on chest radiographs when compared with conventional TBB.

Epidemiological evidence shows that as many as 22% of patients with IPF develop lung cancer; this risk is nearly five times greater than that of the general population [16]. However, the most successful diagnostic approaches to PPLs in these patients are unestablished. The most recent American Thoracic Society/European Respiratory Society/Japanese Respiratory Society/Latin American Thoracic Society guidelines (updated in 2018) do not account for the diagnostic effect of IPF on PPLs [17]. In this study, the diagnostic yield of malignant lesions in patients with IPF was higher using EBUS-GS, although the difference was not significant. A recent meta-analysis reported that the weighted diagnostic yield for malignant lesions by R-EBUS was approximately 60–70% [18]. By our findings, the diagnostic yield of EBUS-GS for malignant lesions in patients with ILD was comparable to what has been previously reported.

Other studies have demonstrated the overall complication rate of EBUS-GS to be 1.3% (13/965). In this regard, 0.8% (8/965) of patients experienced pneumothorax and 0.5% (5/965) experienced subsequent pulmonary infection [19]. In this study, the complication rate of EBUS-GS in patients with ILD was slightly higher than has been previously documented (approximately 5%). However, when compared with the frequency of pneumothorax that occurs with TTNB [20], we believe that EBUS-GS offers an acceptable alternative for the initial diagnosis of PPLs in patients with ILD.

Our study was limited by the small number of consecutive patients with ILD and by its retrospective design. Larger sample sizes and a prospective study design are needed to confirm and further explore our observations.

## 5. Conclusions

In conclusion, the findings of our study suggest that EBUS-GS is a useful and acceptable initial diagnostic procedure for diagnosing PPLs in patients with ILD. In doing so, our study works at addressing a diagnostic shortfall of lesion assessment in ILD-afflicted lungs with the intent of supporting the early detection of malignant lesions and bettering the outcomes in patients with ILD.

## Figures and Tables

**Figure 1 diagnostics-11-02269-f001:**
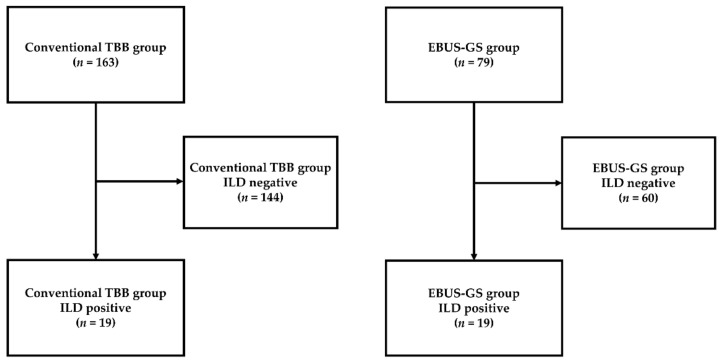
Patient selection and classification flowchart. TBB, transbronchial biopsy; ILD, interstitial lung disease; EBUS-GS, endobronchial ultrasonography with a guide sheath.

**Figure 2 diagnostics-11-02269-f002:**
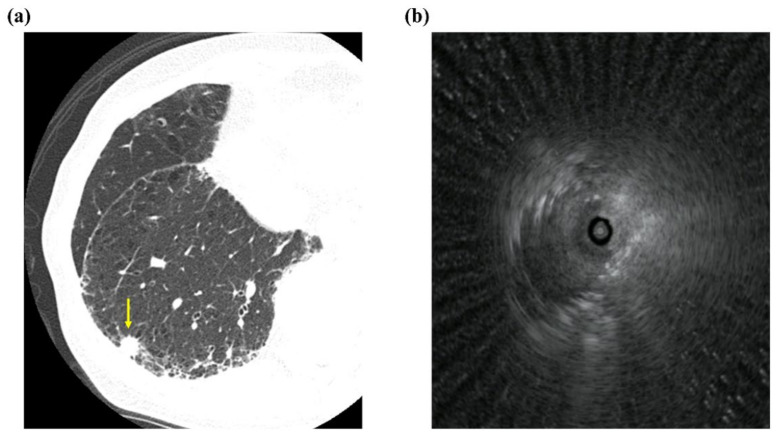
Representative imaging data. (**a**) Chest computed tomography image showing peripheral pulmonary nodule with interstitial lung shadow (arrow), and (**b**) representative radial endobronchial ultrasonography image for within probe position.

**Table 1 diagnostics-11-02269-t001:** Comparison of patient and peripheral pulmonary lesion characteristics between the conventional transbronchial biopsy (TBB) group and the EBUS-GS group (TBB using endobronchial ultrasonography with a guide sheath).

	TBB (*n* = 19)	EBUS-GS (*n* = 19)	*p*-Value
Age in years, median (range)	67.5 (49–79)	73.7 (67–80)	0.194
Male patients, *n* (%)	17 (89.4)	15 (78.9)	0.330
IPF, *n* (%)	7 (36.8)	5 (26.3)	0.364
FVC, percent predicted (%), median (range)	91.9 (59–133)	91.7 (38–130)	0.739
DL_CO_, percent predicted (%), median (range)	64.3 (32–92)	74.5 (41–133)	0.600
Size (mm), median (range)	25.9 (8.0–42.6)	23.4 (15.5–38.8)	0.194
Structure, solid, *n* (%)	19 (100)	18 (94.7)	0.500
Lesion lobe			0.199
Right upper/left upper	5 (26.3)	9 (47.4)	
Right middle/lingula	2 (10.5)	0 (0)	
Right lower/left lower	12 (63.2)	10 (52.6)	
Lesion location, *n* (%)			0.328
Outer	9 (47.4)	12 (63.2)	
Bronchus sign, *n* (%)			0.447
Positive	16 (84.2)	13 (68.4)	
Radiographic visibility, *n* (%)			
Visible	17 (89.4)	13 (68.4)	0.116
Sampling biopsy, *n*, median (range)	5.1 (2–10)	6.2 (3–10)	0.194
Total examination time (min), median (range)	21.1 (10–42)	29.2 (10–52)	0.129

IPF, idiopathic pulmonary fibrosis; FVC, forced vital capacity; DL_CO_, diffusing capacity of the lung for carbon monoxide.

**Table 2 diagnostics-11-02269-t002:** Summary of diagnostic yield according to lesion size, lobe, location, and bronchus sign when comparing lesions assessed using conventional transbronchial biopsy (TBB) or TBB using endobronchial ultrasonography with a guide sheath (EBUS-GS).

		TBB (*n* = 19)	EBUS-GS (*n* = 19)	*p*-Value
Size, *n* (%)	≤20 mm	0/7 (0)	5/7 (71.4)	0.010
	>20 mm	4/12 (33.3)	7/12 (58.3)	0.273
Lobe, *n* (%)	Upper	3/9 (33.3)	3/5 (60.0)	0.329
	Middle/lingula	0/0 (0)	2/2 (100)	-
	Lower	1/10 (10.0)	7/12 (58.3)	0.022
Location, *n* (%)	Outer	1/9 (11.1)	7/12 (58.3)	0.067
	Inner	3/10 (30)	5/7 (71.4)	0.153
Bronchus sign, *n* (%)	Positive	3/16 (18.8)	9/13 (69.2)	0.006
	Negative	1/3 (33.3)	3/6 (50)	0.635
Visibility on chest X-ray, *n* (%)	Visible	3/17 (17.6)	9/13 (69.2)	0.002
	Invisible	1/2 (50)	3/6 (50)	0.500
Total, *n*/19 (%)		4/19 (21.1)	12/19 (63.2)	0.009

**Table 3 diagnostics-11-02269-t003:** Diagnoses based on either conventional transbronchial biopsy (TBB) or TBB using endobronchial ultrasonography with a guide sheath (EBUS-GS).

		TBB (*n* = 19)	EBUS-GS (*n* = 19)	*p*-Value
Malignant lesions	Total diagnostic	3	8	0.053
	Adenocarcinoma	0	1	
	Squamous cell carcinoma	3	5	
	Small cell carcinoma	0	1	
	Non-small cell carcinoma	0	1	
	Total non-diagnostic	11	6	
Benign lesions	Total diagnostic	1	4	0.103
	Inflammatory lesion	0	3	
	Non-tuberculous mycobacteria	1	0	
	Microscopic polyangiitis	0	1	
	Total non-diagnostic	4	1	

**Table 4 diagnostics-11-02269-t004:** Complications when comparing lesions assessed using conventional transbronchial biopsy (TBB) or TBB using endobronchial ultrasonography with a guide sheath (EBUS-GS).

	TBB (*n* = 19)	EBUS-GS (*n* = 19)	*p*-Value
Complications, all	2 (10.5%)	1 (5.3%)	0.451
Pneumothorax, *n* (%)	0 (0%)	1 (5.3%)	-
Bleeding, *n* (%)	2 (10.5%)	0 (0%)	-

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
