# Peer review of "A Pilot Study of Transbronchial Biopsy Using Endobronchial Ultrasonography with a Guide Sheath in the Diagnosis of Peripheral Pulmonary Lesions in Patients with Interstitial Lung Disease"

_diagnostics, 2021, doi:10.3390/diagnostics11122269_

Round 1
Reviewer 1 Report
Dear Authors,
Manuscript is very interesting. I have a few concerns about this paper.
1, Title could be more precise. My suggestion: A pilot study of EBUS-GS TBB in the diagnosis of peripheral pulmonary lesions in ILD patients. I think it is acceptable to use EBUS-GS TBB and ILD if this is a common term in the field.
2, I would recommend to present complication rates data in a table. I think this is very important as you proposed to widely use this technique in all ILD patients. Table presentations will be more straightforward.
Author Response
Comment 1: Title could be more precise. My suggestion: A pilot study of EBUS-GS TBB in the diagnosis of peripheral pulmonary lesions in ILD patients. I think it is acceptable to use EBUS-GS TBB and ILD if this is a common term in the field.
Response: We would like to thank Reviewer for the time and effort in reviewing our manuscript and providing comments and suggestions, which have considerably helped us improve our manuscript. We have answered each of your comments below and hope that our responses and revisions have addressed all of them.
In accordance with the Reviewer’s comment, we changed the manuscript title to ‘A pilot study of EBUS-GS TBB in the diagnosis of peripheral pulmonary lesions in patients with ILD’, as we considered that the expression “patients with ILD” was preferable to “ILD patients”.
Comment 2: I would recommend to present complication rates data in a table. I think this is very important as you proposed to widely use this technique in all ILD patients. Table presentations will be more straightforward.
Response: Based on your suggestion, we have added a table providing data on the complication rates for the TBB and EBUS-GS groups as Table 4 in the revised manuscript.
Reviewer 2 Report
The work idea is great. The scientific work submitted for publication is very important for clinical practice. The test method is appropriate. However, there are doubts about the interpretation of some of obtained results.
The interpretation and presentation of the „Benign lesions“ results in Table 3 are questionable. A histological finding such as „Inflammatory lesion“ may be unrelated (in my opinion, certainly unrelated) to the isolated peripheral lesion. Such a finding may be due to a concomitant interstitial lung disease (ILD). It is necessary to provide information about concomitant IPL for every of these cases.
The „Microscopic polyangiitis“ (MPA) finding is also certainly unrelated to the isolated peripheral lesion. MPA is not characterized by an isolated peripheral focus. In addition, MPA does not have a specific histological manifestation in the lungs.
„Non-tuberculous mycobacteria“ as a finding also cannot be considered as an expression of an isolated peripheral lung lesion. Almost certainly NTM-induced lung disease is a concomitant IPL.
Given that the interpretation of the „Benign lesions“ results is questionable, it is very likely that the statistical significance of the EBUS-GS will disappear in the final result.
In general, this does not reduce the high value of the work, but the presentation, interpretation and final conclusion of the results need to be corrected.
Author Response
Comment 1: The interpretation and presentation of the „Benign lesions“ results in Table 3 are questionable. A histological finding such as „Inflammatory lesion“ may be unrelated (in my opinion, certainly unrelated) to the isolated peripheral lesion. Such a finding may be due to a concomitant interstitial lung disease (ILD). It is necessary to provide information about concomitant IPL for every of these cases.
Response: We would like to thank Reviewer for the time and effort in reviewing our manuscript and providing comments and suggestions, which have considerably helped us improve our manuscript. We have answered each of your comments below and hope that our responses and revisions have addressed all of them.
We corrected the title of Table 3 by deleting the term "histological findings". In our study, bronchoscopic diagnosis of benign lesions was defined depending on whether a definitive diagnosis was obtained histologically and/or microbiologically based on the bronchoscopy-based findings and clinical features—successful diagnosis of benign lesions using bronchoscopy was defined when the specific findings (e.g., granuloma) were confirmed via bronchoscopy and the lesions regressed spontaneously over the follow-up period; the lesions were then diagnosed as inflammatory lesions. In the revised manuscript, we have specified our definition of benign/inflammatory lesions in patients with interstitial lung disease (ILD) more clearly.
Comment 2: The “Microscopic polyangiitis“ (MPA) finding is also certainly unrelated to the isolated peripheral lesion. MPA is not characterized by an isolated peripheral focus. In addition, MPA does not have a specific histological manifestation in the lungs.
Response: In our study, there were some patients had multiple pulmonary nodules in one patient. However, biopsy was performed for one PPL each in 19 patients with ILD who underwent TBB, as well as in another 19 patients with ILD who underwent EBUS-GS. In patients diagnosed with MPA by bronchoscopy, the pathological findings of the specimen obtained by TBB showed infiltration of inflammatory cells around small vessels, and MPA was diagnosed based on the pathological findings of TBB and the patient’s clinical features. We have now mentioned how benign lesions such as those in MPA were diagnosed based on bronchoscopy and clinical features in the revised manuscript.
Comment 3: “Non-tuberculous mycobacteria“ as a finding also cannot be considered as an expression of an isolated peripheral lung lesion. Almost certainly NTM-induced lung disease is a concomitant IPL.
Response: In our study, there were some patients had multiple pulmonary nodules in one patient. However, biopsy was performed for one PPL each in 19 patients with ILD who underwent TBB, as well as in another 19 patients with ILD who underwent EBUS-GS. In the patient diagnosed with non-tuberculous mycobacteria (NTM) infection based on conventional TBB, mycobacterial histologic feature of granulomatous inflammation and positive culture for NTM were confirmed. In addition to these histological and microbiological bronchoscopy findings, chest CT findings in the patient were consistent with those of most patients with NTM. Therefore, we diagnosed NTM infection. In the revised manuscript, we have additionally mentioned that in patients who underwent TBB or EBUS-GS, biopsy was performed for one PPL each and have also described the definition of benign lesions that was used more clearly.
Comment 4: Given that the interpretation of the „Benign lesions“ results is questionable, it is very likely that the statistical significance of the EBUS-GS will disappear in the final result.
Response: We think that the statistical significance of the results for EBUS-GS compared to those for TBB is valid, because we dealt with benign lesions based on definite diagnostic criteria, which have now been more clearly explained in the revised manuscript.
Round 2
Reviewer 2 Report
Yes, the manuscript has been sufficiently improved for publication in Diagnostics.